# Integrating Aggressive-Variant Prostate Cancer-Associated Tumor Suppressor Gene Status with Clinical Variables to Refine Prognosis and Predict Androgen Receptor Pathway Inhibitor Response in Metastatic Hormone-Sensitive Setting

**DOI:** 10.3390/ijms26115309

**Published:** 2025-05-31

**Authors:** Martino Pedrani, Giuseppe Salfi, Sara Merler, Irene Testi, Chiara Maria Agrippina Clerici, Giovanna Pecoraro, Luis Castelo-Branco, Fabio Turco, Luigi Tortola, Ursula Vogl, Silke Gillessen, Jean-Philippe Theurillat, Thomas Zilli, Ricardo Pereira Mestre

**Affiliations:** 1Oncology Institute of Southern Switzerland (IOSI), Ente Ospedaliero Cantonale (EOC), 6500 Bellinzona, Switzerland; martino.pedrani@eoc.ch (M.P.); giuseppe.salfi@eoc.ch (G.S.); chiaramariaagrippina.clerici@eoc.ch (C.M.A.C.); fabio.turco@eoc.ch (F.T.); luigi.tortola@eoc.ch (L.T.); ursula.vogl@eoc.ch (U.V.); silke.gillessen@eoc.ch (S.G.); 2Department of Oncology and Hemato-Oncology, Università degli Studi di Milano, 20100 Milan, Italy; 3Institute of Oncology Research (IOR), 6500 Bellinzona, Switzerland; jean-philippe.theurillat@ior.usi.ch; 4Faculty of Biomedical Sciences, Università della Svizzera Italiana, 6900 Lugano, Switzerland; thomas.zilli@eoc.ch; 5Section of Innovation Biomedicine–Oncology Area, Department of Engineering for Innovation Medicine, University of Verona and Verona University Hospital Trust, 37129 Verona, Italy; 6Medical Oncology Unit, University Hospital of Parma, 43126 Parma, Italy; 7Oncology Department, University of Naples “Federico II”, 80131 Naples, Italy; giovanna.pecoraro@eoc.ch; 8Department of Radiation Oncology, Oncology Institute of Southern Switzerland, Ente Ospedaliero Cantonale, 6500 Bellinzona, Switzerland; 9Faculty of Medicine, University of Geneva, 1206 Geneva, Switzerland

**Keywords:** prostate cancer, aggressive-variant prostate cancer, tumor suppressor genes, *TP53*, *RB1*, *PTEN*/*PI3K*/*AKT*, androgen receptor pathway inhibitors, survival, prognostic, predictive

## Abstract

Alterations in aggressive-variant prostate cancer-associated tumor suppressor genes (AVPC-TSG: *TP53*, *RB1*, *PTEN*) are related with androgen insensitivity and aggressive disease. However, their prognostic and predictive role in metastatic hormone-sensitive prostate cancer (mHSPC) is unclear. This single-center retrospective study assesses the value of AVPC-TSG alterations in refining prognosis and predicting the response to androgen receptor pathway inhibitors (ARPIs) in mHSPC. We included 158 patients with genomic tumor sequencing undergoing treatment for mHSPC between 2013 and 2023. We compared patients with AVPC-TSGalt tumors (≥1 alteration in *TP53*, *RB1*, or *PTEN*/*PI3K*/*AKT* pathway genes) to those with AVPC-TSGwt tumors (i.e., without alterations in AVPC-TSG). Cox analyses were performed for progression-free survival (PFS) and overall survival (OS). AVPC-TSGwt status was associated with improved PFS and OS in both univariate and multivariate (MV) analyses (MV PFS: HR 0.58, 95%CI: 0.38–0.89, *p* = 0.012; MV OS: HR 0.48, 95%CI: 0.26–0.91, *p* = 0.025). AVPC-TSGalt mHSPC patients seemed to derive no PFS benefit from ARPI addition (PFS: HR 1.13, 95%CI: 0.58–2.19, *p* = 0.721), while AVPC-TSGwt mHSPC patients did (PFS: HR 0.51, 95%CI: 0.28–0.93 *p* = 0.029). Integrating AVPC-TSG status with CHAARTED volume criteria, we identified three distinct subgroups: “good risk” (AVPC-TSGwt low volume), “intermediate risk” (either AVPC-TSGalt low volume or AVPC-TSGwt high volume), and “poor risk” (AVPC-TSGalt high volume) with median PFS of 46.8, 28.2, and 15.7 months, respectively. Only the “intermediate risk” subgroup seemed to derive PFS benefit from ARPI addition (HR 0.36, 95%CI: 0.19–0.70, *p* = 0.002). AVPC-TSG status assessment refines prognosis and may predict PFS benefits of ARPIs in mHSPC. AVPC-TSGalt mHSPC patients should be considered for clinical trials as they may not benefit from current standard approaches.

## 1. Introduction

Prostate cancer (PC) is the second most commonly diagnosed solid tumor and the fifth leading cause of mortality in men worldwide [1]. While treatment options for metastatic hormone-sensitive prostate cancer (mHSPC) have expanded and combination therapies have been shown to prolong survival [2], it is of interest to identify patients who could benefit from less intensive treatment regimens to avoid unnecessary toxicity.

Current clinical parameters for predicting progression-free survival (PFS) and overall survival (OS) in mHSPC are inadequate to fully capture the multifaceted and heterogeneous nature of the disease. The STOPCAP meta-analysis incorporating CHAARTED volume criteria with type of disease presentation (de novo or metachronous) showed that patients with metachronous and low-volume disease experience long survival and do not benefit from the addition of docetaxel. However, in patients with high-volume diseases, 5-year PFS and OS rates were similar regardless of disease presentation, indicating a lack of effective long-term prognosis discrimination in this group [3].

Genomic profiling has underscored the significance of tumor suppressor gene (TSG) alterations in PC biology, which has led to the definition of aggressive-variant PC (AVPC), a particular form of metastatic castration-resistant PC (mCRPC). AVPC is molecularly defined by combined defects in *TP53*, *RB1*, and *PTEN* (AVPC-TSG), and this genetic signature may predict a favorable response to platinum-based therapies [4].

On the other hand, AVPC-TSG alterations have also been associated with poor prognosis and androgen insensitivity [5]. The loss of *TP53*, *RB1*, and *PTEN* drives rapid disease progression and hormonal therapy resistance, a phenomenon well established in preclinical studies [6,7,8,9,10,11], with supporting clinical evidence primarily derived from the mCRPC setting [12,13,14].

While the acquisition of AVPC-TSG alterations has been linked to previous hormonal therapy exposure, these alterations are frequently found in hormone-naïve samples as well [15]. Their predictive and prognostic impact may therefore vary depending on the stage of PC, the timing of their assessment, previous treatment exposure, and the presence of isolated or combined defects. This results in a lack of treatment recommendations in clinical PC guidelines based on the presence or absence of these alterations.

In patients with mHSPC treated with androgen deprivation therapy (ADT) and androgen receptor pathway inhibitors (ARPIs), ancillary studies from the STAMPEDE platform utilizing transcriptome-wide analysis identified *PTEN* and *TP53* loss as negative prognostic factors [16]. Additionally, a single-institution retrospective study demonstrated that AVPC-TSG altered status (AVPC-TSGalt), defined as the presence of any alteration in one or more AVPC-TSG detected through genomic tumor sequencing, was associated with significantly shorter PFS in patients receiving ADT plus abiraterone compared to those with AVPC-TSG wild-type (AVPC-TSGwt) mHSPC (8.0 months vs. 23.2 months, respectively) [17].

However, these studies did not evaluate the impact of AVPC-TSG alterations on risk stratification when integrated with clinical variables and did not compare treatment outcomes across different regimens (e.g., ADT monotherapy, ADT + docetaxel, or ADT + ARPI) and other molecular subgroups.

As a result, it remains unclear whether all patients with AVPC-TSGalt mHSPC derive meaningful PFS or OS benefits from ARPI or triplet therapy. Furthermore, the interaction between AVPC-TSG status and established clinical prognostic variables has not been fully explored, leaving their role in refining prognosis and guiding treatment decisions in mHSPC unclear.

Our study aims to address this knowledge gap by evaluating the prognostic value of AVPC-TSG status in a treatment-naïve setting and its predictive role in determining the benefit of ARPI in patients with mHSPC. These findings aim to enhance personalized clinical decision-making, particularly in identifying patients who may benefit from treatment de-escalation or escalation strategies.

## 2. Results

### 2.1. Study Population Characteristics

The ESMO-GROW flowchart for real-world evidence studies in oncology is presented in sResults1. During the inclusion period (January 2013–December 2023), 158 mHSPC patients met the inclusion criteria. Table 1 presents the baseline demographics and clinical variables of the overall population at the time of treatment initiation, distinguishing between those presenting with AVPC-TSGalt and those with AVPC-TSGwt mHSPC.

Within the study population, 100 (63.3%) patients presented with de novo mHSPC, 88 (55.7%) had high-volume disease (accordingly to CHAARTED criteria), and 80 (50.4%) had ISUP Grade Group 5 disease. Notably, 7 (4.4%) presented with liver metastasis and 18 (11.4%) with lung metastasis. The overall median PFS was 28.2 months (95%CI: 23.8–39.6; range: 1–129 months), and the median OS was 87.5 months (95%CI: 68.2-NR; range: 1–129 months).

The analyzed tissue samples were derived from radical prostatectomies, prostate biopsies, or transurethral resection of the prostate specimens in 74% of cases, and biopsy of a local or distant metastatic site (lymph nodes, liver, peritoneum, bone, lung, soft tissue) in the remaining cases (Appendix A).

In the overall population, 63 (39.9%) patients presented at least one altered AVPC-TSG (*TP53* in 29.7%, *RB1* in 1.9%, and *PTEN*/*PI3K*/*AKT* pathway in 12.7%), with 7 (4.4%) patients presenting combined AVPC-TSG alterations. Type and frequency of detected alterations are described in Appendix A.

When comparing patients with AVPC-TSGalt versus AVPC-TSGwt tumors, no differences were observed in median age, median PSA at mHSPC diagnosis, or traditional clinical prognostic variables (CHAARTED volume criteria, ISUP Grade Group, disease presentation). Conversely, lung metastases were more common in AVPC-TSGalt tumors (19.9% vs. 6.3%). Notably, the type of mHSPC treatment did not significantly differ between AVPC-TSGalt and AVPC-TSGwt tumors (*p* = 0.64). Among the 58 patients with metachronous mHSPC, 28 (48.3%) received pelvic radiotherapy and 22 (37.9%) ADT in the localized setting.

### 2.2. Impact of AVPC-TSG Alteration on Survival Outcomes

The effects of AVPC-TSG status on PFS and OS from univariate and multivariate Cox analysis including traditional clinical variables (CHAARTED volume criteria, ISUP Grade Group, type of disease presentation) are presented in Table 2 and Table 3. Forest plots for PFS and OS MV Cox analysis are shown in Appendix A.

In the overall population, we found AVPC-TSGalt status (defined as the presence of at least one AVPC-TSG alteration) to be an independent negative prognostic factor in both univariate and multivariate Cox analysis for PFS (AVPC-TSGwt versus AVPC-TSGalt: HR: 0.51, 95%CI: 0.29–0.92, *p* = 0.025; HR: 0.58, 95%CI: 0.38–0.89, *p* = 0.012, respectively) and OS (AVPC-TSGwt versus AVPC-TSGalt: HR: 0.47, 95%CI: 0.26–0.87, *p* = 0.017; HR: 0.48, 95%CI: 0.26–0.91, *p* = 0.015). Median PFS and OS were significantly shorter in the AVPC-TSGalt mHSPC population compared to AVPC-TSGwt (PFS: 20.5 versus 39.6 months; OS: 68.2 months versus not reached with a lower 95%CI of 94.6 months) as depicted by the Kaplan–Meier curves in Figure 1A,B. Tests for interaction between AVPC-TSG status and clinical variables yielded negative results (Table 2 and Table 3).

We conducted further MV Cox analysis for PFS and OS including number and site of metastasis (lung or liver metastasis) (Appendix A). AVPC-TSG status remained significantly associated with PFS and OS (AVPC-TSGwt versus AVPC-TSGalt: HR: 0.56, 95%CI: 0.36–0.87, *p* = 0.010; HR: 0.43, 95%CI: 0.22–0.84, *p* = 0.014, respectively). Liver metastasis was negatively associated with PFS and OS (HR: 4.27, *p* < 0.001; HR: 4.21, *p* = 0.018), while a positive association was found for oligometastatic disease presentation (HR: 0.58, *p* = 0.088; HR: 0.27, *p* = 0.041). ISUP Grade Group and type of disease presentation were significantly associated with OS in the MV Cox analysis (*p* = 0.017, *p* = 0.050). Conversely, CHAARTED volume criteria were not associated with PFS and OS in the MV Cox analysis (*p* = 0.480; *p* = 0.962). Tests for interaction between AVPC-TSG status and clinical variables were all negative (Appendix A). Figure 1C,D show the forest plots for PFS and OS MV Cox analysis. Importantly, when sensitivity analyses (MV Cox analysis performed with AVPC-TSGalt status and each clinical variable singularly) were performed, AVPC-TSGalt status retained prognostic significance across all models (Appendix A).

The effect of AVPC-TSGwt status on patient survival outcomes was similar in de novo and metachronous mHSPC, both in terms of PFS and OS, though not always reaching statistical significance (de novo mHSPC PFS: HR: 0.53 *p* = 0.020; de novo mHSPC OS HR: 0.53 *p* = 0.096; metachronous mHSPC PFS: HR: 0.61, *p* = 0.169; metachronous mHSPC OS: HR: 0.47, *p* = 0.185) (Appendix A). Patients whose tumors presented with two or more concomitant AVPC-TSG alterations (n = 7) showed a trend towards a higher risk of progression and death compared to those with only one alteration (n = 56), as shown in Appendix A. However, the small sample size limits definitive conclusions.

### 2.3. Refining CHAARTED Criteria Including AVPC-TSG Status

AVPC-TSG status was integrated with CHAARTED clinical criteria to create a new variable with three levels:-“AVPC-TSGwt/LV” (absence of AVPC-TSG alterations and presence of low-volume disease).-“AVPC-TSGalt/HV” (presence of both AVPC-TSG alterations and high-volume disease).-“AVPC-TSGalt/LV or AVPC-TSGwt/HV” (presence of either AVPC-TSG alteration or high-volume disease).

Cox analyses for PFS and OS were conducted together with pairwise comparisons. Subgroup comparison through Holm’s method found statistically significant differences between all three subgroups in terms of PFS (*p* < 0.05), with a similar trend in terms of OS (Appendix A).

We identified a “good risk subgroup” (“AVPC-TSGwt/LV”) with patients having exceptional PFS and OS, a “poor risk subgroup” with poor survival outcomes (“AVPC-TSGalt/HV”) (median PFS: 46.8 versus 15.7 months; median OS: not reached versus 62.3), and an “intermediate risk subgroup”, as depicted by the Kaplan–Meier curves in Figure 2. Notably, type of mHSPC treatment did not significantly differ across the three subgroups (Appendix A). No PFS differences were noted when comparing patients with “AVPC-TSGalt/LV” to those with “AVPC-TSGwt/HV” mHSPC (Appendix A).

Similarly, we integrated AVPC-TSG status with type of disease presentation and ISUP Grade Group. Cox analyses for PFS and OS, along with pairwise comparisons, are presented in Appendix A. Corresponding Kaplan–Meier curves are shown in Appendix A. Conversely, the integration of clinical variables alone (disease volume, ISUP Grade Groups, and disease presentation) failed to effectively discriminate PFS outcomes when AVPC-TSG alterations were not considered (Appendix A).

### 2.4. Predictive Value of AVPC-TSG Alteration Status

Univariate Cox analyses for PFS based on the use of an ARPI-based regimen (ADT + ARPI or ADT + ARPI + docetaxel) in the overall population and in AVPC-TSGalt and AVPC-TSGwt subgroups are shown in Appendix A. Kaplan–Meier PFS curves are depicted in Figure 3. Only the AVPC-TSGwt subgroup derived a PFS benefit from the addition of an ARPI to mHSPC treatment (HR: 0.51, 95%CI: 0.28–0.93, *p* = 0.029). No PFS difference was observed for patients with AVPC-TSGalt mHSPC treated with ARPI-based regimes compared to ADT ± docetaxel (HR: 1.13, *p* = 0.072)

Exploratory univariate Cox analysis for PFS based on the use of ARPI-based regimens in mHSPC patients, stratified by the newly established prognostic risk groups, is detailed in Appendix A. Kaplan–Meier PFS curves are illustrated in Figure 3. Only the intermediate risk subgroup (“AVPC-TSGalt/LV or AVPC-TSGwt/HV”) showed a statistically significant PFS benefit from ARPI-based regimens (HR: 0.36, 95%CI: 0.19–0.70, *p* = 0.002), particularly evident in the “AVPC-TSGwt/HV” subgroup (Appendix A).

### 2.5. Quality Assessment

The quality and completeness of the real-world data analysis was evaluated considering the ESMO-GROW guidance and self-reported informative score (Appendix A).

## 3. Discussion

To the best of our knowledge, this is the first study to examine the impact of AVPC-TSGalt status (assessed through commercially available next-generation sequencing platform) on refining the prognostic value of renowned clinical variables for the risk of progression and death in patients with mHSPC. It is also the largest study exploring the role of AVPC-TSGalt status in predicting PFS benefits from ARPI-based treatment regimens (ADT + ARPI or ADT + ARPI + docetaxel).

We found that the presence of at least one AVPC-TSG alteration was associated with shorter PFS and was an independent prognostic factor for OS, with significant and clinically relevant effect sizes in both univariate and multivariate analyses. Patient groups (AVPC-TSGalt vs. AVPC-TSGwt) were well balanced in terms of clinical prognostic factors and treatments received. Despite the limited sample size, sensitivity analyses corroborated our findings, with AVPC-TSG status maintaining statistical significance in all multivariable Cox regression models. Additionally, no interaction was found between AVPC-TSG status and clinical variables, further supporting its independent prognostic role.

We developed a new prognostic stratification of mHSPC patients by integrating AVPC-TSG status with CHAARTED volume criteria, resulting in three distinct risk groups with significantly different PFS estimates. The “poor risk group”, characterized by high-volume disease and AVPC-TSG alterations, had the worst prognosis (median PFS: 15.7 months). Conversely, nearly 50% of the “good risk group”, with low-volume disease and no AVPC-TSG alterations, were free from progression or death at 4 years (median PFS: 46.8 months), with over 70% surviving at 5 years. Notably, patients with either a genomic poor-risk feature (AVPC-TSGalt/LV) or a clinical high-risk feature (AVPC-TSGwt/HV) exhibited similar PFS and OS, defining the intermediate risk group—a decision made a priori and supported by these comparable outcomes and pragmatic sample size considerations. Treatment type in mHSPC, particularly ARPI-based regimens, was evenly distributed across subgroups. In addition, the integration of clinical variables among them (disease volume, ISUP Grade Groups, and disease presentation) failed to effectively discriminate PFS outcomes when AVPC-TSG alterations were not taken into account. Our findings highlight the prognostic value of AVPC-TSG status in mHSPC, enhancing risk stratification and survival predictions.

While ARPI-based regimens (ADT + ARPI or ADT + ARPI + docetaxel) outperform ADT monotherapy or ADT + docetaxel in randomized clinical trials [3,18], treatment decisions in the mHSPC setting rely solely on clinical variables, with triplet therapy preferred in patients with high-volume de novo disease [19,20]. However, ARPI toxicities including financial burden [21] warrant consideration. The impact of AVPC-TSG status on treatment outcomes remains unclear and is not factored into treatment decision-making in this setting.

As expected, in our study, ARPI-based regimens improved PFS overall (HR = 0.64, *p* = 0.044). However, only AVPC-TSGwt patients benefited significantly (HR = 0.51, *p* = 0.029), while AVPC-TSGalt tumors showed no advantage (HR = 1.13, *p* = 0.721).

The intermediate risk subgroup showed the greatest benefit from ARPI-based strategies (HR = 0.36, *p* = 0.002), even exceeding the overall AVPC-TSGwt population. Detailed analysis revealed that the PFS benefit was primarily driven by patients with high-volume/AVPC-TSGwt disease, reinforcing AVPC-TSG alterations as negative predictors of ARPI response. In contrast, neither the good risk (HR = 0.98, *p* = 0.959) nor poor risk (HR = 1.07, *p* = 0.873) subgroups showed significant benefit from ARPI-based treatment. The absence of a clear advantage in these groups may also reflect limited sample size and heterogeneity in treatment regimens. However, very few patients in these subgroups received docetaxel.

Moreover, the absence of ARPI benefit aligns with both preclinical studies and mCRPC clinical data which indicate that AVPC-TSG alterations may mediate resistance to hormonal treatments and docetaxel, with *TP53*, *RB1*, and *PTEN* loss driving prostate cancer progression and ARPI resistance.

Mutant p53 compromises docetaxel response [22] and fails to downregulate AR, promoting resistance [6,7,8]. *RB1* loss enables castration-resistant tumor growth [9,10,11] and resistance to abiraterone/enzalutamide [23,24,25,26], while *TP53*/*PTEN* co-loss is linked to abiraterone resistance [27]. However, clinical evidence in mHSPC remains inconclusive.

Clinically, *TP53* alterations predict a poorer response [12] and shorter PFS and OS [13] to abiraterone and enzalutamide in mCRPC. Similarly, *PTEN* loss is associated with a shortened response to abiraterone [14]. STAMPEDE transcriptomic analysis identified *TP53*/*PTEN* loss as a negative prognostic factor in mHSPC patients treated with ARPI [16], confirmed by a retrospective study linking AVPC-TSGalt status to inferior PFS with ADT plus abiraterone [17]. A multicenter retrospective analysis further showed that the survival benefit from ADT plus docetaxel was limited to the TSGwt group, with no significant differences in the TSGlow counterpart [28].

Our analysis, supported by the aforementioned preclinical and clinical research, suggests that patients in the “AVPC-TSGwt/LV” subgroup (61.4% of the low-volume population) might be ideal candidates for de-escalation strategies, as their long survival estimates are likely due to high sensitivity to prostate cancer treatments, a favorable intrinsic phenotype, or both. Additionally, specific genomic alterations in AVPC-TSGwt tumors (such as SPOP mutations) may further enhance their sensitivity to hormonal treatments [29]. Patients in the intermediate risk group (50% of the study population) may benefit from ARPI-based regimens and standard treatment intensification. Conversely, patients with both high-volume and AVPC-TSG-altered mHSPC (40.9% of the high-volume population) have a very poor prognosis and should be considered for clinical trials exploring alternative treatments, as they may not benefit from current standard approaches. Notably, within this high-risk population, those with *PTEN*-altered tumors may be future candidates for *AKT* inhibitors, as a recent press release from the CAPItello-281 trial reported significant radiographic PFS improvement with capivasertib in *PTEN*-deficient mHSPC [15]. Triplet therapy (ADT + ARPI + docetaxel) is mostly considered for high-volume mHSPC, but AVPC-TSG alterations may confer resistance to both docetaxel and ARPIs. Thus, assessing its efficacy in relation to the AVPC-TSG mutational status will be crucial to avoid overtreatment, unnecessary toxicities, and increased mortality.

Notably, a study employing whole-genome sequencing (WGS) and whole-transcriptome sequencing (WTS) of biopsies from ARPI-treated mCRPC patients found that *TP53*, *PTEN*, and *RB1* aberrations did not correlate with treatment response, suggesting that prior treatments may affect molecular marker reliability [30]. Conversely, our study using treatment-naïve biopsies eliminates such confounders, ensuring a more unbiased evaluation of AVPC-TSG alterations.

Additionally, AVPC-TSG status varies depending on testing timing and site due to spatial and temporal heterogeneity [31]. Hamid et al. reported TSG variants in 39% of localized prostate cancer cases, rising to 63% in mHSPC and 92% in mCRPC [32]. In our cohort, considering *PI3K*/*AKT* pathway-associated genes, 39.9% had at least one altered AVPC-TSG, likely due to 74% of NGS analyses having been conducted on primary prostate specimens. This highlights the prognostic significance of AVPC-TSG alterations before treatment and metastasis.

Given the limited surgical options and tissue availability in these patients, NGS analysis on primary tumors offers a practical alternative for real-world clinical applications. Our study utilized commercially available platforms, proving that routine genomic analyses can provide valuable prognostic and predictive insights if further validated in a prospective study. Furthermore, our study included a real-world patient cohort treated at a tertiary referral center according to standard treatment and staging protocols, enhancing the applicability of our findings to routine oncology practices.

The main limitations of our study include its retrospective single-center nature, with a limited sample size and treatment heterogeneity, which prevented us from testing the effect of individual treatment strategies across subgroups. These are common limitations when assessing retrospective data from real-world practice. To mitigate these issues, patients included were derived from a consecutive series, and we conducted multivariable analysis considering all relevant clinical variables, as well as sensitivity analyses. Notably, our genomic analyses were performed on a single mHSPC sample per patient, overlooking potential intratumoral heterogeneity. Moreover, our NGS panel did not assess gene deletions, a frequent TSG alteration. Notably, these are typical limitations of clinical-grade NGS panels, which we employed in line with our goal to make our findings reproducible and cost-effective in most clinical settings. Our study lacks functional validation and presents potential confounders (comorbidities, socioeconomic factors). We finally underscore the need for larger datasets and prospective studies to validate and extend the insights gained from our study.

## 4. Materials and Methods

The analysis of this real-world evidence study has been reported in accordance with ESMO Guidance for Reporting Oncology real-World Evidence (ESMO-GROW) recommendations [33].

This is a single-center retrospective study analyzing data extracted from electronic medical records of patients who consecutively presented with mHSPC and initiated systemic therapy at the Oncology Institute of Southern Switzerland between the years 2013 and 2023 with either ADT alone or in association with docetaxel and/or ARPI, who also had tumor tissue genomic testing performed on specimens collected prior to ADT initiation or had archival treatment-naïve tissue available for retrospective analysis. Clinical-grade next = generation sequencing (NGS) of FFPE tissue from biopsies of patients with metastatic disease was performed by Ion Torrent (Thermo Fisher Scientific, Waltham, MA, USA) using Oncomine Comprehensive Assay v3 (Thermo Fisher Scientific, Waltham, MA, USA). Oncomine Comprehensive Assay v3 is a multi-biomarker NGS assay that covers 161 cancer-related genes, including *TP53*, *PTEN*, *AKT*, PIK3C/R, and *RB1*, all of interest in this study.

We focused on *TP53*, *RB1*, and *PTEN* alterations because each has well-documented preclinical evidence of driving resistance to androgen receptor pathway inhibitors, and all three are readily and reproducibly assayed by clinical NGS panels.

Every patient either signed an informed consent form for data collection or was deceased at the time of data analysis. The study was in the scope of a retrospective data collection protocol approved by the local ethical committee and was in accordance with the 1964 Helsinki Declaration and its later amendments or comparable ethical standards. Details on inclusion criteria, exclusion criteria, and genomic sequencing analysis can be found in sMethod1–3. PFS and OS definitions [34,35] are detailed in sMethod4. Several demographic and clinical parameters were gathered with genomic data, as shown in sMethod4–5.

Our main objectives were to evaluate whether integrating AVPC-TSG status with established prognostic factors—CHAARTED high-volume criteria, metastatic presentation (de novo vs. metachronous), and ISUP Grade—could refine traditional risk stratification and predict benefit from ARPIs in mHSPC. Our primary and secondary objectives are described in detail in sMethod6 together with exploratory analysis.

Pearson’s chi-square test was employed to examine the association between categorical variables and the Kruskal–Wallis rank sum test was used for continuous measures. AVPC-TSGalt status was defined as the presence of one or more alterations in the *TP53* gene, *RB1* gene, or genes involved in the *PTEN*/*PI3K*/*AKT* pathway, as detected by genomic tumor sequencing. Conversely, AVPC-TSGwt status was characterized by the absence of any such alterations. We conducted comparative analysis between patients with AVPC-TSGalt and AVPC-TSGwt mHSPC. PFS and OS were estimated with the Kaplan–Meier method, and group comparisons were made using the log-rank test. All statistical comparisons were made with two-tailed tests. Multivariable Cox regression analysis for PFS and OS and a test of interaction were performed to assess the independent prognostic value of AVPC-TSG status alongside demographic and clinical variables (CHAARTED volume criteria, de novo/metachronous presentation, ISUP Grade Group) [36].

To provide a more comprehensive analysis and pinpoint patients at very high or very low risk of progression and death, we integrated genetic risk (AVPC-TSG status) with clinical risk, classifying patients with both features as high risk, those with neither as low risk, and those with only one feature as intermediate risk. Therefore, AVPC-TSG status was incorporated into relevant clinical variables as detailed in sMethod7, and Holm’s correction method was used to compare subgroups [37].

The results are presented as hazard ratios (HRs) with 95% confidence intervals (95%CIs). All statistical analyses were carried out using R statistical software version 4.4.2 and Jamovi statistical software version 2.5.6. A more detailed explanation of the conducted statistical analysis, including exploratory and sensitivity analysis, can be found in sMethod8–9.

## 5. Conclusions

Our findings reveal for the first time that AVPC-TSG status provides additional prognostic and, potentially, predictive information beyond traditional clinical variables in mHSPC. AVPC-TSG status evaluation can reshape mHSPC prognosis and inform treatment selection, identifying a subgroup of patients with exceptional survival outcomes—those with AVPC-TSGwt and low-volume disease—who may be potential candidates for de-escalation strategies. At the same time, patients with AVPC-TSGalt mHSPC should be considered for clinical trials exploring alternative treatments, as they may not benefit from current standard approaches. Accounting for AVPC-TSG mutational status will be important in future mHSPC trials for developing more personalized treatment strategies, including intensification and de-intensification of available modalities and testing innovative treatments.

## Figures and Tables

**Figure 1 ijms-26-05309-f001:**
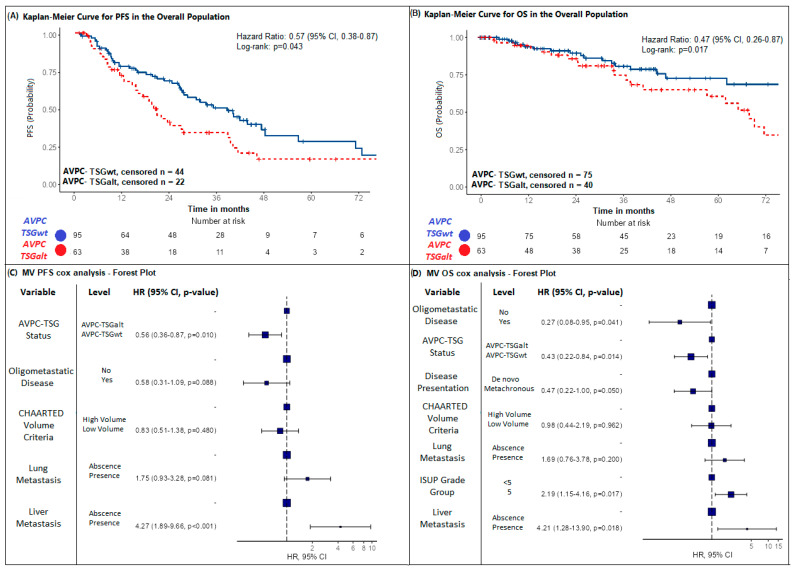
Kaplan–Meier curves and forest plots for progression-free survival (PFS) and overall survival (OS) in the overall population. (**A**) Kaplan–Meier curves resulting from UV PFS Cox regression analysis comparing patients with AVPC-TSGalt and AVPC-TSGwt mHSPC. (**B**) Kaplan–Meier curves resulting from UV OS Cox regression analysis comparing patients with AVPC-TSGalt and AVPC-TSGwt mHSPC. (**C**) A forest plot resulting from MV PFS Cox regression analysis including AVPC-TSG status and all clinical variables shown to be statistically significant in the UV Cox regression analysis (shown in Appendix A). (**D**) A forest plot resulting from the MV OS Cox regression analysis including AVPC-TSG status and all clinical variables shown to be statistically significant in the UV Cox regression analysis (shown in Appendix A). Abbr. PFS = progression-free survival; OS = overall survival; UV = univariate; MV = multivariate; Met = metastasis; AVPC-TSG = aggressive-variant cancer-associated tumor suppressor gene; mHSPC = metastatic hormone-sensitive prostate cancer.

**Figure 2 ijms-26-05309-f002:**
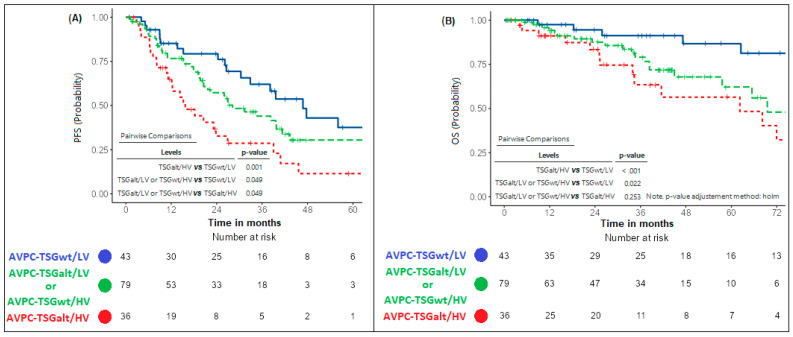
Progression-free survival (**A**) and overall survival (**B**) by integrating AVPC-TSG alteration status and CHAARTED volume criteria. Kaplan–Meier curves resulting from UV PFS (**A**) and OS (**B**) Cox regression analysis comparing patients with AVPC-TSGalt tumors and CHAARTED high-volume disease (“TSGalt/HV”), with AVPC-TSGwt tumors and CHAARTED low-volume disease (“TSGwt or LV”), and with either AVPC-TSGalt tumors or CHAARTED high-volume disease (“TSGalt/LV or TSGwt/HV”). Abbr. PFS = progression-free survival; OS = overall survival; UV = univariate; AVPC-TSG = aggressive-variant cancer-associated tumor suppressor genes; mHSPC = metastatic hormone-sensitive prostate cancer; HV = high volume; LV = low volume.

**Figure 3 ijms-26-05309-f003:**
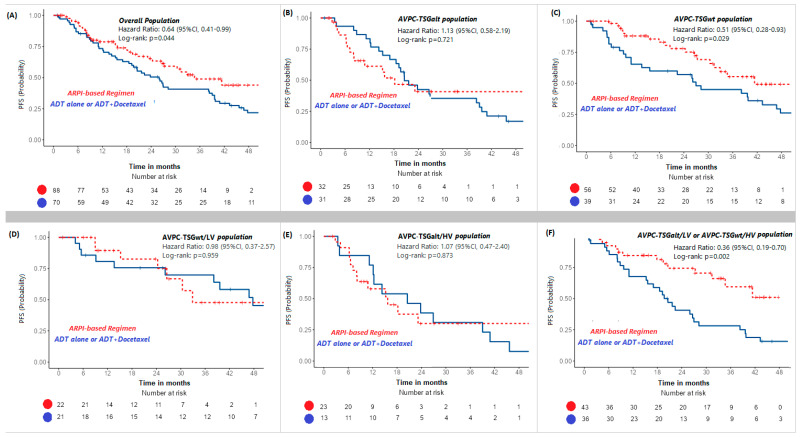
Progression-free survival by mHSPC treatment in the overall population and in different patients subgroups. Kaplan–Meier curves resulting from UV PFS Cox regression analysis comparing mHSPC treatment with ARPI-based regimens (ADT + ARPI or ADT + ARPI + Docetaxel) versus treatment with ADT alone or ADT + Docetaxel in the overall population (**A**) and in patients with AVPC-TSGalt tumors (**B**), AVPC-TSGwt tumors (**C**), CHAARTED low-volume disease and the absence of AVPC-TSG alterations (AVPC-TSGwt/LV) (**D**), CHAARTED high-volume disease and the presence of AVPC.TSG alterations (AVPC-TSGalt/HV) (**E**), or the presence of either high-volume disease or AVPC-TSG alterations (AVPC-TSGalt/LV or AVPC-TSGwt/HV) (**F**). Abbr. PFS = progression-free survival; UV = univariate; AVPC-TSG = aggressive-variant cancer-associated tumor suppressor gene; mHSPC = metastatic hormone-sensitive prostate cancer; ADT = androgen deprivation therapy, ARPI = androgen receptor pathway inhibitor; HV = high volume; LV = low volume.

**Table 1 ijms-26-05309-t001:** Patient baseline demographics comparing patients with AVPC-TSGalt vs AVPC-TSGwt tumors.

Prognostic Variable		All Patients n = 158 (%)	TSG Alt	TSG wt	*p* Value
			n = 63 (39.9%)	n = 95 (60.1%)	
**Median age, years (Min, Max)**		73 (47, 91)	72.8 (54, 89)	73.1 (47, 91)	*p* = 0.72 ^1^
**Age—categorical**					*p* = 0.53 ^2^
	<75 yo	93 (58.9%)	39 (61.9%)	54 (56.8%)	
	≥75 yo	65 (41.1%)	24 (38.1%)	41(33.2%)	
**Median PSA (Min-Max)**					*p* = 0.961
		29.1 [Min: 0.038–Max: 6231]	28 [Min: 0.770–Max: 5609]	30 [Min: 0.038–Max: 6231]	
**PSA at mHSPC diagnosis**					*p* = 0.96 ^2^
	PSA < 10	36 (22.8%)	15 (23.8%)	21 (22.1%)	
	PSA 10–100	86 (55.1%)	34 (54%)	53 (55.8%)	
	PSA ≥ 100	35(22.2%)	14 (2.2%)	21 (22.1%)	
**mHSPC treatment**					*p* = 0.64 ^2^
	ADT alone	55 (34.8%)	23 (36.5%)	32 (33.6%)	
	ADT + ARPI	77 (48.7%)	28 (44.5%)	49 (51.6%)	
	ADT + Docetaxel	15 (9.5%)	8 (12.7%)	7 (7.4%)	
	ADT + ARPI + Docetaxel	11 (7%)	4 (6.3%)	7 (7.4%)	
**De Novo or Relapsed**					*p* = 0.29 ^2^
	De Novo	100 (63.3%)	43 (68.3%)	57 (60%)	
	Relapsed	58 (36.7%)	20 (31.7%)	38 (40%)	
**Chaarted Volume**					*p* = 0.77 ^2^
	High Volume	88 (55.7%)	36 (57.1%)	52 (54.7%)	
	Low Volume	70 (44.3%)	27 (42.9%)	43 (45.3%)	
**ISUP grade**					*p* = 0.77 ^2^
	<5	78 (49.4%)	32 (50.1%)	46 (48.4%)	
	5	80 (50.6%)	31 (49.9%)	49 (51.6%)	
**Oligometastatic disease**					*p* = 0.55 ^2^
	No	129 (81.6%)	50 (79.4%)	79 (83.2%)	
	Yes	29 (18.4%)	13 (20.6%)	16 (16.8%)	
**Bone met**					*p* = 0.14 ^2^
	No	38 (24.1%)	19 (30.2%)	19 (20%)	
	Yes	120 (75.9%)	44 (69.8%)	76 (80%)	
**Liver met**					*p* = 0.53 ^2^
	No	151 (95.6%)	61 (96.8%)	90 (94.7%)	
	Yes	7 (4.4%)	2 (3.2%)	5 (5.3%)	
**Lung met**					*p* = 0.01 ^2^
	No	140 (88.6%)	51(80.1%)	89 (93.7%)	
	Yes	18 (11.4%)	12 (19.9%)	6 (6.3%)	
***PTEN*/*PI3K*/*AKT* pathway alteration status**					
	*PTEN*/*PI3K*/*AKT* wt	138 (87.3%)	43 (68.3%)	95 (100%)	
	*PTEN*/*PI3K*/*AKT* alt	20 (12.7%)	20 (31.7%)	0(0%)	
***RB1* alteration status**					
	*RB1* wt	155 (98.1%)	60 (95.2%)	95 (100%)	
	*RB1* alt	3 (1.9%)	3 (4.8%)	0 (0%)	
***TP53* alteration status**					
	P53 wt	111 (70.3%)	16 (25.4%)	95 (100%)	
	P53 alt	47 (29.7%)	47 (74.6%)	0(0%)	
**Number of AVPC-TSG alterations**					
	AVPC-TSG wt	95 (60.1%)	0(%)	95 (100%)	
	AVPC-TSG 1 alt	56 (35.4%)	56 (88.9%)	0 (0%)	
	AVPC-TSG 2–3 alt	7 (4.4%)	7 (11.1%)	0 (0%)	
**Prostate RT in met setting**					*p* = 0.74 ^2^
	No	131 (82.9%)	53 (84.1%)	78 (82.1%)	
	Yes	27 (17.1%)	10 (25.9%)	17 (17.9%)	
**median PFS (months Min–Max)**		28.2 (IC95% 23.8–39.6) [Min: 1.13–Max: 129]	20.5 (IC95% 15.7–38.3)	39.6 (IC95% 28.2–56.1)	*p* = 0.010 ^3^
**PFS-censored patients**		66 (41.8%)	22 (34.9%)	44 (46.3%)	
**median follow-up for PFS censored (months 95%CI)**		40.7 (IC95% 33.7–45.2) [Min: 1.00–Max: 104]	33.7 (IC95% 27.3-NA)	41.2 (IC95% 34.2–45.2)	
**median OS (months Min–Max)**		87.5 (IC95% 68.2-NR) [Min: 3.90–Max: 128]	68.2 (IC95% 57.6-NR)	NR (IC95% 94.6-NR)	*p* = 0.017 ^3^
**OS-censored patients**		115 (72.8%)	40 (63.5%)	75 (78.9%)	
**median follow-up for OS censored (months 95%CI)**		41(34.2–44.6) [Min: 1–Max: 129]	72.5 (60.9-not reached)	80.5 (59.9-not reached)	

^1^ Wilcoxon. ^2^ Pearson. ^3^ Log rank test. Abbreviations: AVPC-TSG = aggressive-variant prostate cancer-associated tumor suppressor gene; PFS = progression-free survival; OS = o survival; CI = confidence interval; PSA = Prostate-Specific Antigen; ADT = androgen deprivation therapy; ISUP = I nternational Society of Urological Pathology; ARPI = androgen receptor pathway inhibitor;; mHSPC = metastatic hormone-sensitive prostate cancer.

**Table 2 ijms-26-05309-t002:** Univariate and multivariate Cox proportional hazard models for PFS.

Prognostic Variable	Levels	Univariate Analysis	Multivariate Analysis
Total N. 158		HR (95%CI), *p*-Value	HR (95%CI), *p*-Value
**Chaarted Volume**	High Volume	-	-
	Low Volume	0.57 (0.37–0.88) *p* = 0.012 **	0.58 (0.37–0.90) *p* = 0.014 **
**De Novo/Metacronous**	De novo	-	-
	Metacronous	0.89 (0.58–1.36) *p* = 0.576	
**AVPC-TSG status**	AVPC-TSGalt	-	-
	AVPC-TSGwt	0.57 (0.38–0.87) *p* = 0.010 **	0.58 (0.38–0.89) *p* = 0.012 **
**ISUP Grade**	<5	-	-
	5	1.22 (0.43–3.44) *p* = 0.703	
**Age at mHSPC**	<75	-	-
	≥75	0.87(0.57–1.32) *p* = 0.506	
**Test for interaction**			
**Chaarted Volume * TSG status**	**Df, Chi-square, *p*-value** 1, 0.0013, 0.971		

Abbreviations: PFS = progression-free survival; CI = confidence interval; HR = hazard ratio; ISUP = International Society of Urological Pathology; Df = degree of freedom; AVPC-TSG = aggressive-variant prostate cancer-associated tumor suppressor gene; mHSPC = metastatic hormone-sensitive prostate cancer. Prognostic variables were included in the multivariable model if *p*-value ≤ 0.10 (*). The MV model in the overall population included CHAARTED volume and TSG status. Prognostic variables included in the multivariable model were deemed statistically significant if *p*-value ≤ 0.05 (**).

**Table 3 ijms-26-05309-t003:** Univariate and multivariate Cox proportional hazard models for OS.

Prognostic Variable	Levels	Univariate Analysis	Multivariate Analysis
Total N. 158		HR (95%CI), *p*-Value	HR (95%CI), *p*-Value
**Chaarted Volume**	High Volume	-	-
	Low Volume	0.37 (0.19–0.72) *p* = 0.003 **	0.49 (0.24–1.01) *p* = 0.052
**Disease Presentation**	De novo	-	-
	Metacronous	0.40 (0.21–0.77) *p* = 0.004 **	0.59 (0.29–1.23) *p* = 0.158
**AVPC-TSG status**	AVPC-TSGalt	-	-
	AVPC-TSGwt	0.47 (0.26–0.87) *p* = 0.017 **	0.48 (0.26–0.91) *p* = 0.025 **
**ISUP Grade**	<5	-	-
	5	1.76 (0.95–3.25) *p* = 0.072 *	2.10 (1.11–3.94) *p* = 0.022 **
**Age at mHSPC**	<75	-	-
	≥75	0.96 (0.52–1.78) *p* = 0.9	
**Test for interaction**			
**Chaarted Volume * TSG status**	**Df, Chi-square, *p*-value** 1, 0.9338, 0.3339		
**Disease Presentation * TSG status**	**Df, Chi-square, *p*-value** 1, 0.2731, 0.6013		
**ISUP Grade * TSG status**	**Df, Chi-square, *p*-value** 1, 0.0118, 0.9134		

Abbreviations: OS = overall survival (from treatment start to death); CI = confidence interval; HR = hazard ratio; ISUP = International Society of Urological Pathology; AVPC-TSG = aggressive-variant prostate cancer-associated tumor suppressor gene; mHSPC = metastatic hormone-sensitive prostate cancer. Prognostic variables were included in the multivariable model if *p*-value ≤ 0.10 *. The MV model in the overall population included CHAARTED volume, liver and lung metastasis, oligometastatic status, TSG status, disease presentation, and ISUP Grade. Prognostic variables included in the multivariable model were deemed statistically significant if *p*-value ≤ 0.05 (**).

## Data Availability

Individual patient data are unavailable due to privacy and ethical restrictions.

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
