# Peer review of "Integrating Aggressive-Variant Prostate Cancer-Associated Tumor Suppressor Gene Status with Clinical Variables to Refine Prognosis and Predict Androgen Receptor Pathway Inhibitor Response in Metastatic Hormone-Sensitive Setting"

_ijms, 2025, doi:10.3390/ijms26115309_

Round 1
Reviewer 1 Report
Comments and Suggestions for Authors
Please refer to the attached reviewer report

Author Response
Reviewer 1.
This is a well-written and clinically meaningful manuscript that provides valuable real-world evidence on the prognostic and predictive role of AVPC-TSG alterations in metastatic hormone-sensitive prostate cancer (mHSPC). Integrating genomic and clinical variables into refined risk groups is a strength, and statistical analyses very well support the findings. That said, I believe the manuscript could benefit from a few refinements:
- The current title is clear but overly long and wordy. It could be trimmed to emphasize better integrating genomic data, predicting ARPI benefit, and identifying distinct prognostic subgroups.
- While the methodology is comprehensive, key elements (e.g., inclusion criteria, genomic testing platform) are entirely deferred to the supplementary section. A summary in the main text would enhance accessibility for readers.
- The Kaplan-Meier and forest plots are valuable, but could be made more intuitive. For example, improving color coding, legends, and group labels (e.g., AVPC-TSGalt/HV) would allow for better standalone interpretation of figures.
- The rationale behind combining AVPC-TSGalt/LV and AVPC-TSGwt/HV into a single intermediate-risk group would benefit from further discussion, including whether this reflects biological equivalence or a pragmatic stratification.
- The manuscript could briefly expand on the translational implications of AVPC-TSG testing, including emerging trials or potential therapeutic avenues targeting these alterations.
- While the limitations are fairly addressed, additional comments on the lack of functional validation and real-world confounders (e.g., comorbidities, socioeconomic factors) could strengthen the rigor of the discussion.
- The manuscript focuses exclusively on TP53, RB1, and PTEN alterations to define AVPC-TSG status. While this is consistent with established definitions of aggressive variant prostate cancer and clinically relevant pathways of androgen insensitivity, it would strengthen the study to clarify whether other genomic alterations associated with ARPI resistance—such as AR amplifications or splice variants, SPOP mutations, MYC, or FOXA1—were assessed but excluded, or not included by design. A brief rationale for focusing solely on these three TSGs would help contextualize the biomarker strategy and distinguish biological versus methodological constraints. Additionally, the rationale for including AKT/PIK3C alterations under 'PTEN pathway' should be explicitly stated in the main text to align with the supplementary material.
The manuscript offers a valuable contribution to prostate cancer genomics. It is scientifically robust and appropriate for publication, subject to minor revisions
Response 1.
Dear Reviewer,
Thank you for your positive assessment and for emphasizing the clinical relevance and robustness of our work. We have carefully addressed each of your suggestions and believe these revisions have significantly strengthened the manuscript:
- We agree that the original title is long. However, we felt it was essential to highlight both the presence of tumor suppressor gene alterations associated with aggressive variant prostate cancer and our novel focus on their role in the hormone-sensitive setting. Even the abbreviated terms “aggressive variant prostate cancer” and “androgen-receptor pathway inhibitors” lengthen the title considerably, so we made only minimal edits to improve clarity and brevity. We hope the revised title now reads more effectively.
- We completely agree and have added a concise summary of key methodological elements to the main Methods section:
- Inclusion criteria
- Genomic testing platform
- Definitions of tumor suppressor gene alterations
- We have standardized figure labels and color schemes across all Kaplan–Meier and forest plots. Please note that the images you saw in the full text were compressed and did not reflect true texture quality; we will provide high-resolution figures upon resubmission.
- From the outset, our study design aimed to separate genomic risk (alterations associated with aggressive variants) from clinical risk (eg. high-volume disease). This created an intermediate-risk category for patients with only one adverse feature. We performed separate analyses of these two subgroups (see Supplementary Table 11) and observed no significant differences in progression-free or overall survival. However we aknoledge it might also be due to the limited sample size. We have now better described it in the methods section and we have expanded the discussion to explain that combining these subgroups was a predefined decision, supported by similar outcomes in our cohort and pragmatic sample-size considerations.
- We fully agree that exploring the translational implications of AVPC-TSG testing and emerging targeted strategies is both important and timely. For the sake of brevity and focus, we have kept this section concise and direct readers to the comprehensive review in reference 15 for a deeper exploration. To underscore a concrete and immediate example, we have added a mention of the recent CAPItello-281 press release, which reported a significant improvement in radiographic progression-free survival with the AKT inhibitor capivasertib in PTEN-altered metastatic hormone-sensitive prostate cancer. Thank you for this valuable suggestion.
- We have enriched our limitations paragraph to acknowledge the lack of functional validation, potential confounders (comorbidities, socioeconomic factors), and the retrospective design.
- We chose to concentrate on TP53, RB1, and PTEN alterations because they form a clinically reproducible signature with well-characterized roles in resistance to androgen-receptor pathway inhibitors. These alterations are supported by extensive preclinical data, especially in the castration-resistant setting, and our goal was to test their prognostic/predictive value in the hormone-sensitive context. Rarer alterations were excluded due to low prevalence and limited sample size. We now explicitly state our rationale for grouping AKT/PIK3C alterations under the PTEN pathway and direct readers to Supplementary Table S2 for the complete genomic profile.
Thank you again for your insightful feedback, which has markedly enhanced the clarity, accessibility, and translational impact of our manuscript.
Sincerely,
Reviewer 2 Report
Comments and Suggestions for Authors
Pedrani et al provide a very interesting manuscript examining the prognostic value of aggressive variant genomic alterations in mHSPC. The rationale for the study is clearly present and of high interest to the community. They provide a single center retrospective analysis to show that CHAARTED risk groups can be further defined with status of mutations in tumor suppression genes to better prognosticate in mHSPC. Overall this is a very interesting and value study that generates a very interesting hypothesis that needs to be tested.
Specific comments:
- Some figures are very small and text is not readable when printed. For example, figure 3 has extremely small text and should be revised for readability.
- The analysis of whether patients with AVPC benefit from early treatment intensification with ARPIs is of high interest. Provocative data is presented in Figure 3 presents data comparing "ARPI-based regimens" vs "ADT alone or ADT+ docetaxel." The conclusions presented are that ARPI may not benefit LV or AVPCmut/HV cases. However, given that the control is either ADT or ADT+docetaxel, how does that affect this? We know from STAMPEDE that comparisons between ADT + docetaxel and ADT + abiraterone (without docetaxel) did not show OS differences for those contemporaneously randomized between those 2 interventions (but did have a small PFS benefit). It is not too surprising to see comparable results here especially with the relatively small numbers in the study. It may be underpowered to detect the differences or may be as the authors state -- that there is maybe no benefit to ARPIs. What percentage of the "control" group got docetaxel and what percentage of the "ARPI" group got triplet therapy?
- I appreciate the enthusiasm for the hypothesis of lack of ARPI benefit but needs more discussion of study limitations, heterogeneity of docetaxel use etc.
- Caution with statements regarding that the AVPC signature being "predictive" here, given that this study is retrospective, it at best is associated with lack of benefit. Future prospective study is warranted to confirm the negative prognostic value and explore whether it is a predictive biomarker.
Author Response
Reviewer 2.
Pedrani et al provide a very interesting manuscript examining the prognostic value of aggressive variant genomic alterations in mHSPC. The rationale for the study is clearly present and of high interest to the community. They provide a single center retrospective analysis to show that CHAARTED risk groups can be further defined with status of mutations in tumor suppression genes to better prognosticate in mHSPC. Overall this is a very interesting and value study that generates a very interesting hypothesis that needs to be tested.
Specific comments:
- Some figures are very small and text is not readable when printed. For example, figure 3 has extremely small text and should be revised for readability.
- The analysis of whether patients with AVPC benefit from early treatment intensification with ARPIs is of high interest. Provocative data is presented in Figure 3 presents data comparing "ARPI-based regimens" vs "ADT alone or ADT+ docetaxel." The conclusions presented are that ARPI may not benefit LV or AVPCmut/HV cases. However, given that the control is either ADT or ADT+docetaxel, how does that affect this? We know from STAMPEDE that comparisons between ADT + docetaxel and ADT + abiraterone (without docetaxel) did not show OS differences for those contemporaneously randomized between those 2 interventions (but did have a small PFS benefit). It is not too surprising to see comparable results here especially with the relatively small numbers in the study. It may be underpowered to detect the differences or may be as the authors state -- that there is maybe no benefit to ARPIs. What percentage of the "control" group got docetaxel and what percentage of the "ARPI" group got triplet therapy?
- I appreciate the enthusiasm for the hypothesis of lack of ARPI benefit but needs more discussion of study limitations, heterogeneity of docetaxel use etc.
- Caution with statements regarding that the AVPC signature being "predictive" here, given that this study is retrospective, it at best is associated with lack of benefit. Future prospective study is warranted to confirm the negative prognostic value and explore whether it is a predictive biomarker.
Response 2.
Thank you very much for your thoughtful and positive feedback on our manuscript. We greatly appreciate your recognition of the study’s rationale and its potential value to the mHSPC community. Your specific suggestions are invaluable for strengthening our work:
- We have enlarged and improved the readability of Figure 3 (and other small figures) so that all text and data remain clear when printed. Unfortunately, the original images were compressed and uploaded in a way that degraded their texture; we have now replaced them with high-resolution versions in accordance with comments from Reviewer 1 and Reviewer 2.
- We agree that the inclusion of docetaxel could potentially confound our conclusions. However, as shown in sTable 10, only a very small proportion of patients in both the ADT + ARPI subgroup and the ADT alone subgroup (for both LV and AVPC<sup>mut</sup>/HV cases) received docetaxel, with numbers well balanced between groups. We therefore believe that the impact of docetaxel on our analysis is minimal and does not alter our conclusions.
- We have expanded the Discussion to acknowledge the study’s limitations, including the retrospective design, small sample size, and heterogeneity of docetaxel use.
- We have adjusted our language regarding the AVPC signature to clarify that, given the retrospective design, our findings demonstrate an association rather than definitive predictive biomarker status, and we emphasize the need for prospective validation.
Thank you again for your encouragement and constructive recommendations, which have markedly improved the clarity and impact of our manuscript.
Sincerely